# The Influence of Maternal Psychological Manifestations on the Mother–Child Couple during the Early COVID-19 Pandemic in Two Hospitals in Timisoara, Romania

**DOI:** 10.3390/medicina58111540

**Published:** 2022-10-27

**Authors:** Cristina Dragomir, Roxana Popescu, Elena Silvia Bernad, Marioara Boia, Daniela Iacob, Mirabela Adina Dima, Ruxandra Laza, Nicoleta Soldan, Brenda-Cristiana Bernad, Alin Eugen Semenescu, Ion Dragomir, Claudiu Elian Angelescu-Coptil, Razvan Nitu, Marius Craina, Constantin Balaceanu-Stolnici, Cristina Adriana Dehelean

**Affiliations:** 1Doctoral School, ‘Victor Babes’ University of Medicine and Pharmacy, Eftimie Murgu Square, No. 2, 300041 Timisoara, Romania; 2Department II—Microscopic Morphology, Discipline of Cellular and Molecular Biology, ‘Victor Babes’ University of Medicine and Pharmacy, Eftimie Murgu Square, No. 2, 300041 Timisoara, Romania; 3ANAPATMOL Research Center, ‘Victor Babes’ University of Medicine and Pharmacy, Eftimie Murgu Square, No. 2, 300041 Timisoara, Romania; 4Department XII—Obstetrics-Gynecology, Discipline of Obstetrics-Gynecology III, ‘Victor Babes’ University of Medicine and Pharmacy, Eftimie Murgu Square, No. 2, 300041 Timisoara, Romania; 5Department XII—Obstetrics-Gynecology, Discipline of Neonatology and Childcare, ‘Victor Babes’ University of Medicine and Pharmacy, Eftimie Murgu Square, No. 2, 300041 Timisoara, Romania; 6‘Pius Brinzeu’ County Emergency Clinical Hospital, Liviu Rebreanu Bd., No. 156, 300723 Timisoara, Romania; 7Department XIII, Discipline of Infectious Disease, ‘Victor Babes’ University of Medicine and Pharmacy, Eftimie Murgu Square, No. 2, 300041 Timisoara, Romania; 8‘Francisc I. Rainer’ Anthropological Research Center, Romanian Academy, Eroii Sanitari Bd., No. 8, 050474 Bucharest, Romania; 9Department VIII—Neurosciences, ‘Victor Babes’ University of Medicine and Pharmacy, Eftimie Murgu Square, No. 2, 300041 Timisoara, Romania; 10Department of Psychology, West University of Timisoara, Vasile Parvan Bd., No. 4, 300223 Timisoara, Romania; 11Institute for Advanced Environmental Research (ICAM), West University of Timisoara, Vasile Parvan Bd., No. 4, 300223 Timisoara, Romania; 12Individual Family Medical Office, 207440 Ostroveni, Romania; 13Rumanian Academy, Calea Victoriei, No. 125, Sector 1, 010071 Bucharest, Romania; 14‘Francisc I. Rainer’ Institute of Anthropology, Romanian Academy, Eroii Sanitari Bd., No. 8, 050474 Bucharest, Romania; 15Department of Toxicology and Drug Industry, Faculty of Pharmacy, ‘Victor Babes’ University of Medicine and Pharmacy, Eftimie Murgu Square, No. 2, 300041 Timisoara, Romania; 16FARMATOX Research Centre for Pharmaco-Toxicological Evaluation, ‘Victor Babes’ University of Medicine and Pharmacy, Eftimie Murgu Square, No. 2, 300041 Timisoara, Romania

**Keywords:** symptoms of postpartum depression, anxiety as a state and as a trait, child bed-sharing, mother–child couple

## Abstract

Background and objectives: The postpartum maternal physical and psychological state played a fundamental role in the mother–child relationship at the beginning of the COVID-19 pandemic. The aim of the study is to analyze the influence of maternal psychological manifestations on the mother–child couple through three objectives (briefly expressed): (I) Determination of the main acute and chronic conditions of newborns/infants. (II) Verification of the hypothesis of the existence of a link between the following neonatal variables: gestational age, birth weight, number of days of hospitalization, and specific neonatal therapies (oxygen, surfactant, and blood products’ transfusion). (III) Verification of the influence of postpartum maternal psychological status on the mother–child couple through three hypotheses. Materials and methods: This cross-sectional study was conducted in two hospitals in Timișoara, Romania, between 1 March and 1 September 2020, and included 165 mothers and their 175 newborns. Mothers answered the Edinburgh Postnatal Depression Scale, Spielberger’s Inventory of State-Trait Anxiety, and the Collins and Read Revised Adult Attachment Scale. Results: (I) The acute and chronic pathology of the infants in the study group was polymorphic. (II) Large correlations were identified between the following infant variables: gestational age with birth weight, and number of hospitalization days with birth weight, gestational age, and use of blood product transfusion (all *p* < 0.001). (III) (1) State anxiety was the only significant predictor of number of hospitalization days (*p* = 0.037), number of acute disorders (*p* = 0.028), and number of infant chronic diseases (*p* = 0.037). (2) Maternal depressive symptoms were the only predictor of postpartum maternal attachment (*p* = 0.018). (3) Depressive symptoms, state, and trait anxiety were non-significant in all models studied (all *p* > 0.05). Conclusions: Postpartum maternal physical and psychological state plays a fundamental role on the mother–child relationship in the new social and complex family conditions.

## 1. Introduction

An analysis of the state of the physical and psychological health of mothers in relation to the health of their newborns is important for improving postpartum maternal and neonatal care and the family environment.

Postpartum mental health status can be analyzed by the maternal psychological manifestations present after birth, such as the presence and intensity of symptoms of anxiety and depression, and by the changes in the maternal attachment style (in the sense of the mother’s attachment behavior towards the newborn) [1,2].

Postpartum depression occurs during the first 4–6 weeks postpartum, of which symptoms may include: depressed mood, loss of interest in activities, sleep disorders, appetite disorders, loss of energy, a feeling of uselessness, guilt, decreased ability to concentrate, irritability, anxiety, and suicidal thoughts [3,4,5,6,7]. A study conducted in Turkey that included 360 pregnant women and that investigated symptoms of postpartum depression with the Edinburgh Scale Postnatal Depression (EPDS) revealed that the prevalence was 35%, and that a pre-existing mental disorder, a somatic disorder, a depressive disorder in the first trimester of pregnancy, domestic violence during pregnancy, placement of the infant in an incubator, and absence of breastfeeding were predictors of symptoms of postpartum depression [8,9].

Another maternal psychological manifestation is anxiety, for which the estimated rate is 17% immediately after birth and 20% during the first six weeks after birth [10]. A narrative review of 14 studies (44 articles, 2407 women) sought to identify descriptions of women’s cognitive, affective, and somatic experiences related to postpartum anxiety disorders, and found that postpartum anxiety is associated with abnormalities in mother–child attachment, symptoms of postpartum depression, reduced likelihood of breastfeeding, increased risk of infant abuse, delayed cognitive and social development, and a high likelihood of anxiety in children [11].

Anxiety can be analyzed as state anxiety (a temporary, transitory, and subjective emotion) and as trait anxiety (a personality dimension characterized by an individual’s predisposition to worry about life-threatening events) [12,13]. The comorbidity rate of anxiety and symptoms of postpartum depression is 75% [14,15].

The first hour after birth is the most critical hour in the life of a mature newborn and skin-to-skin contact and breastfeeding (recommended for up to 6 months) in the first hour of life are essential for the mother–child relationship and for the whole family [16,17,18,19]. Additionally, the Global Strategy for Women’s, Children’s, and Adolescents’ Health (2016–2030), Strategies toward Ending Preventable Maternal Mortality and Every Newborn Action Plan, considers the postnatal period, which lasts from immediately after the birth of the child until 6 weeks (42 days) after the birth, to be a critical period for women, newborns, partners, parents, caregivers, and families [20].

One study that analyzed the specific risk of transmitting psychopathology from mentally ill parents to offspring noted that the changes related to psychopathology in the behavior and physiology of the parents in the perinatal period are related to the behavioral, biological, and neurophysiological correlations of the psychological functioning of the infant during this period [21]. This highlights the importance of early detection and intervention aimed at parents’ mental disorders to break the chain of the intergenerational transmission of psychopathology [22,23].

Attachment theory pioneer John Bowlby focused on understanding the nature of the infant–caregiver relationship and believed that attachment characterizes human experience throughout life. Later, Hazan and Shaver studied Bowlby’s attachment theory in the context of romantic relationships and concluded that adult romantic relationships, like infant–caregiver relationships, are attachments [24,25]. A study included in the Northern Babies Longitudinal Study (NorBaby), from 2015 to 2017, investigated the attachment style during pregnancy and postpartum depression symptoms in 168 women and proved that the avoidant attachment style is an individual predictor for the mother–child relationship [26]. In other words, the attachment style of adults may be related to parental stress in the first year after birth, and this may lead to parents having a negative relationship with their own child. Many studies have demonstrated the links between the mother’s adult-type attachment in the postpartum period and high anxiety or postpartum depression [1]. Other studies have shown that the mother’s attachment type can be a risk factor for postpartum depression or anxiety [27,28]. Stable vulnerability factors (for example, personal history of depression, anxiety and psychopathology of other family members, personality disorders), specific relationship-related life events (for example, breakups, marriage, divorce, death, giving birth), and changes in underlying perceptions of oneself and others can produce changes in adult-type attachment [29].

At it is understood, the postpartum mother’s physical and psychological health, the newborns’ health, and the mother–child relationship are essential elements of the postpartum period.

Birth can occur prematurely (before 37 weeks of gestation), at term (between 37 and 42 weeks of gestation), and post-term (over 42 weeks of gestation), and can often have some complications [30]. Newborns, depending on birth weight (BW), are classified as micro-preemie—BW < 800 g, extremely low birth weight (ELBW)—BW < 1000 g, very low birth weight (VLBW)—BW < 1500 g, low birth weight (LBW)—BW < 2500 g, normal birth weight (NBW)—2500–4000 g, high birth weight (HBW)—4000–4500 g, and very high birth weight (VHBW)—BW > 4500 g [31].

Neonatal/young infant pathology is diverse, often with origins in intrauterine life, and includes acute conditions and diseases with chronic evolution, such as birth trauma, thermoregulation disorders, hydro-electrolytic disorders, neonatal hyperbilirubinemia, carbohydrate metabolism disorders, respiratory, cardiovascular, hematologic, neurological, and endocrine disorders, inborn errors of metabolism, infections, dermatological and sensory diseases, malformative pathology, conditions requiring surgical interventions, etc. [32,33,34].

Bonding is an emotional bond between the mother and child that develops during pregnancy and is an important component of the mother–child relationship [35]. Type of attachment, postpartum maternal psychological health, socio-demographic factors (for example, family economic level, family type, and mother’s education level), physical health of mothers and newborns/infants, breastfeeding, and other influencing factors enhance the mother–child relationship [36,37].

The World Health Organization’s Every Newborn Action Plan (2014) aimed to accelerate progress towards reducing neonatal deaths and stillbirths and reducing maternal morbidity and mortality [38].

The aim of this study is to analyze the influence of maternal psychological manifestations on the mother–child couple after a brief analysis of the health of mothers (physical and psychological) and newborns.

(I) In this sense, the first objective of the study was to determine the main acute and chronic conditions of newborns/infants hospitalized together with their mothers.

(II) Another objective is to see if the hypothesis of the existence of a link between the following neonatal variables is verified: gestational age (GA), BW, number of days of specialization, and specific neonatal therapies: oxygen therapy (O), surfactant (S), and blood transfusion blood products (T).

(III) Another objective of the present work was to verify the following hypotheses regarding the influence of postpartum maternal psychological status on the mother–child couple:

(1) The same postpartum maternal psychological manifestations can create a predictive pattern regarding the number of hospital days of the mother–child pair, the number of acute conditions, and the number of chronic diseases of the newborn/infant.

(2) To what extent do symptoms of anxiety (state and trait) and intensity of postpartum depression symptoms predict the mother’s type of adult attachment in the postpartum period?

(3) Maternal psychological manifestations will predict sharing of the bed with the child, evaluation of the child, breastfeeding, and the type of nutrition the newborn receives during hospitalization.

## 2. Materials and Methods

This cross-sectional study was performed in the neonatology department at the ‘Pius Brinzeu’ County Emergency Clinical Hospital and the ‘Louis Turcanu’ Emergency Clinical Hospital for Children, Timisoara, from 1 March to 1 September 2020.

In this study, 165 accompanying mothers and their 175 newborns were included (19 twins and triplets). The sample of mothers consisted of 73 primiparous singleton pregnancies, 52 second-parous singleton pregnancies, 7 mothers with first/second birth twins, and 22 multiples (including 1 twin and 1 triplet pregnancy).

Data were selected from clinical observation files during hospitalization.

The criteria of inclusion of mothers were hospitalization with newborns and accurate completion of psychological tests. The criteria for exclusion of mothers were the absence of the mother’s hospitalization with the newborn and the absence of or incomplete response to psychological tests.

The inclusion criterion for newborns or infants was hospitalization in one of the two clinics, accompanied by their mothers. The exclusion criterion was hospitalization of the newborn without the presence of the mother.

Due to strict hospitalization guidelines during the COVID-19 pandemic (WHO), we were unable to have an accurate record of the number of women invited to participate in this study, nor the number of those who declined. Therefore, the number of hospitalized mothers with newborns enrolled in the study was much smaller, and the collection of specific data was clearly limited.

During the hospitalization, after signing the informed consent form, the mothers responded to three psychological tests.

(1) The Edinburgh Postnatal Depression Scale (EPDS) is a screening tool for detecting the risk of postpartum maternal depression. Answers receive 0, 1, 2, and 3 points (direct items) and 3, 2, 1, and 0 points (reverse items). The total score is a maximum of 30 points and measures the intensity of the symptoms of postpartum depression (mother’s condition in the last 7 days, not gest on the day of the test). Depression can be mild/possible (score over 10), moderate to severe (score over 13), and severe (score over 15) [39,40]. The EPDS scale for the antenatal period was validated for the Romanian population and was named the EPDS-Romanian version (EPDS-R) [41]. There are studies in the postnatal period in Romania that used EDPS. For example, two studies from the “Bega” Obstetrics Outpatient Clinic in Timisoara, Romania, detected postpartum depression in 23.93–50% of new mothers, with this condition being significantly correlated with anxiety as a trait [42,43].

Cronbach’s alpha coefficient was used to estimate the reliability (internal consistency) of the test. The Cronbach’s alpha coefficient for the EPDS was 0.84 in this study and this value reflects high reliability (internal consistency) for the scale items. In the specialized literature, Cronbach’s alpha coefficient for the Romanian version of the EDPS (EDPS-R) is 0.88 (very good) [41]. The main limitation of using the EPDS in this study is the assessment of only the intensity of postpartum depressive symptoms in the last 7 days, not a long-term and global assessment of maternal depression.

(2) Spielberger’s Inventory of State-Trait Anxiety (STAI-Y) consists of two scales: a scale for measuring anxiety as a condition, the STAI Y1 form, and a scale for measuring anxiety as a trait, the Y2 form. Each scale has 20 items and an answer scale: 1—not at all, 2—less, 3—moderate, and 4—very much. The total score is obtained on each scale (a maximum of 80 points), and anxiety is divided into mild (35 points or more), moderate (50 points or more), and severe (65 points or more) [12]. STAI-Y is a test validated on the Romanian population, but not specifically in the postnatal period. There are postpartum studies on samples of Romanian women. One example was carried out in university-based obstetrical care units, which showed that state anxiety is more frequently encountered in the postnatal period (33.7% of mothers) compared to the antenatal period (15.5%), and trait anxiety is a predictor for the presence of suicidal ideation [44].

The Cronbach’s alpha coefficient for STAI-Y1 was 0.69 and for STAI-Y2 was 0.62 in this study. In the literature, the Cronbach’s alpha coefficient for STAI-Y1 is 0.93 and for STAI-Y2 is 0.93, and 0.94 and 0.90 for the Romanian inventory [12,45]. Of course, speculations can be made on cultural factors, reluctancy to declare a state of anxiety, or even the unclear understanding of some items by the participants, but we did not want to go into too many details, so as not to divert attention from the purpose the study.

(3) In terms of attachment to the couple’s relationship, the Collins and Read Revised Adult Attachment Scale (AAS) assesses the style of adult attachment in relation to the couple’s partner (we used this test due to the similarity of the attachment with the infant–caregiver relationships). The scale has 18 items: 6 items for each attachment style. Each item was evaluated on a 5-point scale: 1—totally disagree, 2—partially disagree, 3—do not know, 4—partially agree, and 5—totally agree [46]. We then calculated the arithmetic mean of the scores for each of the three attachment styles.

The Cronbach’s alpha coefficient was 0.60 in this study. In the specialty literature, the Cronbach’s alpha coefficient is 0.65 [47]. There are some recent studies on the Romanian population in which the Collins and Read Revised Adult Attachment Scale was used, but without a big relevance to this study [48,49,50,51].

The tests used were self-administered under the supervision of a psychologist, guided by the psychologist on the neonatology ward.

We used SPSS v23 for the statistical analysis of the data. A frequency analysis was calculated for some data on mothers and newborns. Using Pearson’s correlation test, the degree of association between two variables measured on the interval scale was evaluated according to a linear model. The coefficient r can take values between −1 (perfectly negative correlation) and 1 (perfectly positive correlation), and a value of 0 signifies the absence of any correlation between the variables.

Regression models were used to assess the extent to which one or more explanatory variables predict an outcome variable. Logistic regression and multinomial regression were performed to study the relationship between the studied maternal psychological manifestations (symptoms of postpartum depression, anxiety as a state, and anxiety as a trait) as predictor variables and the significance of the regression models [52,53,54].

## 3. Results

### 3.1. Newborns and Mothers

In this study, 116 full-term and 59 premature newborns were included.

In turn, the preterm infants in the study (according to a user-friendly classification) were LBW (*n* = 15), VLBW (*n* = 8), LBW (*n* = 23), and NBW (*n* = 8) (See Table 1).

To understand the investigated postpartum psychological manifestations (intensity of symptoms of postpartum depression and level of symptoms of anxiety as a state and as a trait), we considered the type of family and the type of birth important. For more details, see Table 1.

Neonatal pathology in the study group was diverse. The most common acute neonatal disorders were affections with dermatological manifestations (for example, jaundice, allergic erythema, and edema) in 78.85% of newborns, respiratory disorders in 41.71%, and hydro-electrolytic disorders in 29.14% (Figure 1a). The most common chronic diseases of infants were neurological diseases in 38.85%, hematological diseases in 32.57%, and ophthalmologic diseases in 11.72% (Figure 1b).

Frequently, during pregnancy, pregnant women had amniotic membranes ruptured prematurely in 10.9%, preeclampsia/eclampsia in 9.1%, and imminent abortion in 6.3%.

### 3.2. Correlations between Neonatal Variables

Using Pearson’s test (and continuous and categorical variables), medium and large correlations were observed between all studied neonatal variables (all *p* < 0.001): (a) BW correlated positively with GA (r = 0.883) and the use of special therapies—oxygen therapy (O) (r = 0.532), administration of surfactants (S) (r = 0.690), and blood products transfusions (T) (r = 0.695)—and negatively with the number of acute disorders (r = −0.601) and the number of chronic diseases (r = −0.645). (b) GA correlated positively with the use of O (r = 0.605) and negatively with the number of acute disorders (r = −0.637), number of chronic diseases (r = −0.690), number of days of hospitalization, and use of S (r = −0.737) and T (r = −0.748). (c) The number of acute disorders was associated positively with the number of chronic diseases (r = 0.585), number of days of hospitalization (r = 0.601), and use of therapies O (r = 0.505), S (r = 0.56), and T (r = 0.533). (d) The number of chronic diseases was associated positively with the number of days of hospitalization (r = 0.709) and the use of O (r = 0.526), S (r = 0.583), and T (r = 0.688). (e) The number of days of hospitalization was associated positively with the use of O (r = 0.560), S (r = 0.768), and T (r = 0.799) (See Table 2).

### 3.3. Logistic Regressions and Multinomial Regressions

The mean scores of postpartum maternal psychological manifestations were the following: for symptoms of postpartum depression, 8.64 ± 5.67, for symptoms of state anxiety, 37.22 ± 12.67, and for symptoms of trait anxiety, 35.71 ± 10.60.

State anxiety, trait anxiety, and depression symptoms of the number of days of hospitalization, number of neonatal acute disorders, and number of infant chronic diseases are shown in Table 3. Regarding the number of days of hospitalization, only state anxiety was found to by a significant predictor (β = 0.25, t = 2.10, *p* = 0.037) when maternal depression and trait anxiety were well-controlled (see Table 3). The three predictors explained about 10% of the variability of the number of days of hospitalization (R^2^_adj_ = 0.10).

Regarding the number of acute neonatal disorders, the results were similar to those of previous analyses: the only significant predictor was state anxiety (β = 0.28, t = 2.21, *p* = 0.028), while the three predictors explained about 6% of the variability of the number of acute disorders (R^2^_adj_ = 0.06). Regarding the number of chronic infant diseases, state anxiety was again the only significant predictor (β = 0.26, t = 2.10, *p* = 0.037), while depression and trait anxiety were non-significant. The three predictors explained about 10% of the variability of the number of chronic diseases (R^2^_adj_ = 0.10) (see Table 3).

Maternal attachment (in the sense of attachment behavior after birth) was secure (*n* = 115), avoidant (*n* = 30), and anxious (*n* = 10). The mean score for secure attachment was 3.05 ± 0.73, for avoidant attachment 2.29 ± 0.67, and for anxious attachment 1.90 ± 0.67.

Using multinomial logistic regression, we predicted the type of attachment using the mother’s depression, the mother’s state anxiety, and the mother’s trait anxiety (symptoms of psychological disorders) as predictors. The *p*-value approached 0.5 (Χ^2^(6) = 11.78, *p* = 0.067, Nagelkerke R^2^ = 0.09), and the model correctly predicted 74.2% of the cases. Of the predictors used, only symptoms of depression were a significant predictor (Χ^2^(2) = 8.07, *p* = 0.018), while mother’s state anxiety (Χ^2^(2) = 0.24, *p* = 0.887) and mother’s trait anxiety (Χ^2^(2) = 0.59, *p* = 0.745) were non-significant.

Among those who scored highly in terms of anxiety, child bed-sharing was either normal (*n* = 142) or pathological (*n* = 21). We used a logistic regression model to predict child bed-sharing using the mother’s depression, the mother’s state anxiety, and the mother’s trait anxiety as predictors (symptoms of psychological disorders). The model was non-significant (Χ^2^(3) = 4.69, *p* = 0.196, Nagelkerke R^2^ = 0.05), as were all the studied predictors (i.e., all *p*-values > 0.05).

The evaluation of the child by the mother was favorable (*n* = 149) or not (*n* = 14). Neonatal nutrition was mixed (*n* = 117), milk formula (*n* = 24), or natural (*n* = 21). Mother’s depression, mother’s state anxiety, and mother’s trait anxiety (symptoms of psychological disorders) were investigated as predictors of evaluation of the child (Χ^2^(3) = 15.42, *p* = 0.001, Nagelkerke R^2^ = 0.20), breastfeeding (Χ^2^(3) = 7.21, *p* = 0.066, Nagelkerke R^2^ = 0.06), and neonatal nutrition (Χ^2^(6) = 7.61, *p* = 0.268, Nagelkerke R^2^ = 0.06) in X logistical regression models. These three variables were non-significant in all models studied (i.e., all *p*-values > 0.05).

## 4. Discussion

Regarding the first objective (I) of the paper, we found that the neonatal pathology in the study group was polymorphic. The most common acute neonatal disorders were pathologies with dermatological manifestations, respiratory disorders, and hydro-electrolytic disorders. The most common chronic diseases among the infants were neurological disorders, hematological disorders, and ophthalmologic disorders, which proves consistent with the literature [31,55].

Regarding objective (II), some medium and large correlations were identified between neonatal variables: (a) BW correlated positively with GA and special therapies (O, S, and T) and negatively with the number of acute disorders and number of chronic diseases. (b) GA correlated positively with the use of O and negatively with the number of acute disorders, number of chronic diseases, number of days of hospitalization, and S and T. (c) The number of acute disorders was associated positively with the number of chronic diseases, number of days of hospitalization, and O, S, and T. (d) The number of chronic diseases was associated positively with the number of days of hospitalization and O, S, and T. (e) The number of days of hospitalization was associated positively with O, S, and T [56,57].

Objective (III) was partially argued:

(1) State anxiety was the only significant predictor for the number of days of hospitalization when maternal symptoms of depression and trait anxiety were controlled. Similarly, only state anxiety was a predictor of the number of acute disorders. Regarding the number of infant chronic diseases, state anxiety was again the only significant predictor, and symptoms of depression and trait anxiety were non-significant. We did not find similar studies regarding this the specialty literature.

(2) We predicted the type of postpartum maternal attachment using symptoms of depression, state, and trait anxiety as predictors. Of the predictors used, only the mother’s depression was significant. Recommendations for future studies include further investigation of the studied relationship and further exploration of its possible effect. The study can be expanded with a larger sample to assure a higher statistical significance.

In the specialty literature, a longitudinal study, conducted during the period 2014–2016, carried out in Lebanon (a country with a special culture) on 150 mothers, investigated whether symptoms of postpartum depression influence the mother–child relationship in the first 3 months postpartum [58,59,60]. The results of the study showed that risk factors for the mother–child relationship included a history of depression, high scores on depression tests, and little social support.

Additionally, a study conducted in Sydney, Australia, including 121 mothers voluntarily hospitalized during their first postnatal year in 2017–2018, aimed to examine the mediating effect of emotional disorders on the relationship between insecure maternal attachment style and self-reported attachment to the infant in hospitalized women [61,62,63]. The authors looked at the mothers’ ability to regulate their emotions, which could influence levels of postpartum well-being and help to engage in sensitive care of the baby, thus developing a beneficial mother–child relationship. In this sense, mothers with insecure attachment styles may have major difficulties establishing a normal relationship with their child and in regulating their emotions in the postpartum period.

Another example is a narrative review of papers published in PubMed and PsycINFO that investigated, among other things, postpartum anxiety, and identified its negative effects on breastfeeding, the mother–child relationship, sleep, mental development, and child health [64,65]. Another prospective study of 800 mothers whose infants ranged in age from 0 to 6 months investigated the feeding behavior of the newborn and attempted to develop a scale for assessing postpartum-specific anxiety: feeding behavior was the result of the specific mood during pregnancy as a predictor of the postnatal situation [66,67].

(3) Symptoms of depression and state and trait anxiety were investigated as predictors of child bed-sharing, breastfeeding, and neonatal nutrition in X logistic regression models. These three variables were non-significant in all models studied.

In the literature, a review conducted over the period 2005–2016 using the MEDLINE databases of Ovid, PsycINFO, and the Cochrane Pregnancy and Childbirth Group study registry, included 122 studies and assessed the consequences of untreated postpartum maternal depression on the infant and mother. The outcomes included maternal consequences of postpartum depression (for example, physical and psychological health), infant consequences (for example, anthropometry, physical health, sleep and motor, cognitive, linguistic, emotional, social, and behavioral development), and mother–child interaction (for example, bonding, breastfeeding, and maternal role) [3,68,69,70].

More recently, a multicenter retrospective study conducted in 12 maternity hospitals in Japan investigated postpartum depression with the EPDS in 10,013 women who gave birth at ≥35 weeks of gestation who had undergone perinatal mental health care [71]. The authors attempted to develop machine learning models to predict depression using a conventional logistic regression and four machine learning algorithms based on maternal clinical data. The results were similar to those of conventional regression models, but the incorporation of 2 weeks’ worth of postpartum control data into this model significantly improved the predictive performance.

The negative psychological impact of the COVID-19 pandemic on the mothers in the postnatal period was major, with complex manifestations such as depression and anxiety. Romanian studies have shown an increase in the incidence of postpartum depression during the COVID-19 pandemic, with negative consequences on the mother–child relationship [72,73].

Some studies from China reported a 27.43% prevalence of postpartum depression and a 33% prevalence of anxiety, with an impact on the whole family and the mother–child pair [74,75]. Studies conducted in Canada during the COVID-19 pandemic indicated increased levels of postpartum depression and anxiety symptoms, with negative effects on mother–child bonding [76].

Promoting neonatal and maternal care with the support of healthcare providers and insurance platforms leads to positive results for the community as a whole [77].

Limitations of the study. The COVID-19 pandemic began on 11 March 2020, and in this context, the study began with many restrictions: the expected number of mothers hospitalized with their newborns has decreased considerably, and due to strict hospitalization rules, only mothers hospitalized with their newborns in separate rooms were included in the study; therefore, the mothers’ access to and participation in newborn care was limited. There are limitations related to the validation of psychometric instruments in Romanian samples. EDPS-R is a scale valid only for antenatal depression (in the third trimester of pregnancy), STAI-Y is valid for Romanian woman (but not specifically for the postpartum period), and AAS is not valid for Romanian new mothers.

## 5. Conclusions

Neonatal pathologies in the study group were diverse and involved both acute and chronic conditions. Some large correlations were identified between neonatal variables: GA with BW, and the number of days of hospitalization with BW, GA, and the use of T. State anxiety was the only significant predictor for the number of days of hospitalization, the number of acute disorders, and the number of infant chronic diseases. Symptoms of maternal depression were the only significant predictor for the type of postpartum maternal attachment. The postpartum maternal physical and psychological state plays a fundamental role in the mother–child relationship in new complex material and family conditions, especially in the current social context.

## Figures and Tables

**Figure 1 medicina-58-01540-f001:**
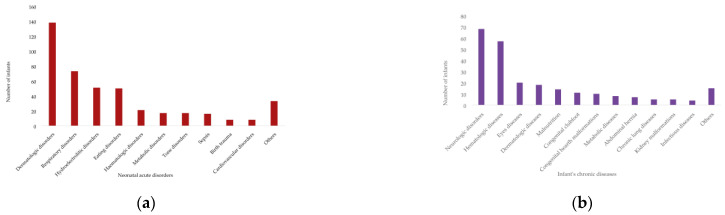
The pathologies of newborns/infants: (**a**) neonatal acute disorders and (**b**) chronic diseases of infants.

**Table 1 medicina-58-01540-t001:** Neonatal and maternal variables.

Variables		*n* (%)
Newborns	Full-term	116 (66.28)
	Premature	59 (33.71)
Premature newborns	Extremely low birth weight	15 (23.45)
	Very low birth weight	8 (13.56)
	Low birth weight	23 (38.98)
	Normal birth weight	9 (15.25)
Oxygen therapy (O)	Free flow in incubator and headbox	122
	Neopuff, ncpap, and/or optiflow	59
	Tracheal intubation and mechanical ventilation	18 (10.29)
Mother’s family	Nuclear	126 (76.36)
	Consensual relationship	18 (10.91)
	Extended	21 (12.72)
Type of birth	Natural	104 (63.03)
	Cesarean section	61 (36.97)

Abbreviation: *n* (%)—number and percentage.

**Table 2 medicina-58-01540-t002:** Correlations between neonatal variables.

Neonatal Variables	1	2	3	4	5	6	7
1. Birth weight	-						
2. Gestational age	0.883						
3. No acute disorders	−0.601	−0.637					
4. No chronic diseases	−0.645	−0.690	0.585				
5. No days of hospitalization	−0.793	−0.788	0.601	0.709			
6. Oxygen therapy (O)	0.532	0.605	0.505	0.526	0.560		
7. Surfactant (S)	0.690	−0.737	0.561	0.583	0.768	−0.377	
8. Blood product transfusion (T)	0.695	−0.748	0.533	0.688	0.799	−0.515	0.648

Notes: *p* < 0.001.

**Table 3 medicina-58-01540-t003:** Coefficients in the linear regression models.

Models 1–3	No. of Days of Hospitalization					No. of Acute Neonatal Disorders					No. of Chronic Infant Diseases				
Variables	R^2^	R^2^_adj_	β	t	*p*	R^2^	R^2^_adj_	β	t	*p*	R^2^	R^2^_adj_	β	t	*p*
Depression	0.11	0.10	0.15	1.46	0.144	0.078	0.061	−0.02	−0.19	0.842	0.12	0.10	0.137	1.28	0.202
Anxiety state			0.25	2.01	0.046			0.28	2.21	0.028			0.26	2.10	0.037
Anxiety trait			−0.04	−0.50	0.613			0.01	0.63	0.950			−0.31	−0.26	0.789

Abbreviations: No.—number; R^2^—coefficient of determination; R^2^_adj_—adjusted coefficient of determination; β—standardized regression coefficient; t—statistic t-test; *p*—statistical significance.

## Data Availability

The data presented in this study are available upon request from the corresponding author.

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
