# Peer review of "The Influence of Maternal Psychological Manifestations on the Mother–Child Couple during the Early COVID-19 Pandemic in Two Hospitals in Timisoara, Romania"

_medicina, 2022, doi:10.3390/medicina58111540_

Round 1

Reviewer 1 Report (Previous Reviewer 2)

The authors of the manuscript made numerous and important corrections and additions. These interventions have certainly improved the configuration of the work, but some further interventions need to be carried out. In particular, (postpartum depression) symptoms / symptomatology is omitted in many parts.

TITLE

The study was carried out to coincide with the first COVID-19 pandemic wave. To facilitate a clearer placement of the scientific article, it would therefore be appropriate that the title also include "... in the first COVID-19 pandemic wave".

ABSTRACT

The abstract must be completely re-written by inserting / omitting the following aspects:

- brief introduction to the investigated area;

- study objectives;

- research period;

- omit the statistical models used and the degrees of significance;

- report only the aspects characterized by the presence of statistical significance.

INTRODUCTION

Page 2, line 81-82: Edinburgh Postnatal Depression Scale;

Page 6.

In the absence of a Romanian validation of the EPDS, some studies that used this tool in the postnatal period in Romania should be reported. Report the EPDS cut-off used in this study.

Likewise, in the absence of a Romanian validation of the STAI, some studies that have used this tool in the postnatal period in Romania should be reported.

Finally, are there any studies with Romanian samples that have the Collins and Read Revised Adult Attachment Scale?

DISCUSSION

Page 12, line 523: some comparisons / considerations should be reported with reference to the results of similar research carried out during these more than two years of the COVID-19 pandemic, both in Romania and in other countries.

LIMITATION OF THE STUDY

It should be noted that the psychometric tools adopted in this study are currently not validated for use in samples of Romanian women.

Author Response

Dear Reviewer,

Thank you for the review and recommendations. We will write down the answer after each of your requests.

Comments and Suggestions for Authors

The authors of the manuscript made numerous and important corrections and additions. These interventions have certainly improved the configuration of the work, but some further interventions need to be carried out. In particular, (postpartum depression) symptoms / symptomatology is omitted in many parts.

Authors:

- We have added the word "symptoms" in the text. Thank you for recommendations.

TITLE

The study was carried out to coincide with the first COVID-19 pandemic wave. To facilitate a clearer placement of the scientific article, it would therefore be appropriate that the title also include "... in the first COVID-19 pandemic wave".

Authors:

- The new title is “The influence of maternal psychological manifestations on the mother-child couple in two hospitals during the early COVID-19 pandemic in Timisoara, Romania”.

ABSTRACT

The abstract must be completely re-written by inserting / omitting the following aspects:

- brief introduction to the investigated area;

- study objectives;

- research period;

- omit the statistical models used and the degrees of significance;

- report only the aspects characterized by the presence of statistical significance.

Authors:

- We rewrote the abstract according to the indications. Thank you for recommendations.

INTRODUCTION

Page 2, line 81-82: Edinburgh Postnatal Depression Scale;

Authors:

- We have corrected the name of the psychological scale.

Page 6.

In the absence of a Romanian validation of the EPDS, some studies that used this tool in the postnatal period in Romania should be reported.

Authors:

- We have completed the text according to the request: ‘’The EPDS scale for the antenatal period was validated for the Romanian population and was named the EPDS-Romanian version (EPDS-R) [42]. There are studies in the postnatal period in Romania that used EDPS. For example, two studies from the "Bega" Obstetrics Outpatient Clinic in Timisoara, Romania, detected postpartum depression in 23.93-50% of new mothers, with this condition being significantly correlated with anxiety as a trait [43,44].”

Report the EPDS cut-off used in this study.

Authors:

- We have completed the text according to the request: ‘’ The main limitation of using the EPDS in this study is the assessment of only the intensity of postpartum depressive symptoms in the last 7 days, not a long-term and global assessment of maternal depression.”

Likewise, in the absence of a Romanian validation of the STAI, some studies that have used this tool in the postnatal period in Romania should be reported.

Authors:

- We have completed the text according to the request: ‘’ STAI-Y is a test validated on the Romanian population, but not specifically in the postnatal period.  There are postpartum studies on samples of Romanian women. One example being carried out in university-based obstetrical care units which showed that state anxiety is more frequently encountered in the postnatal period (33.7% of mothers) compared to the antenatal period (15.5%), and trait anxiety is a predictor for the presence of suicidal ideation [45].’’

Finally, are there any studies with Romanian samples that have the Collins and Read Revised Adult Attachment Scale?

Authors:

- We have completed the text according to the request: ‘’There are some recent studies on the Romanian population in which the Collins and Read Revised Adult Attachment Scale was used, but without a big relevance to this study [49,52].’’ Thank you for recommendations.

DISCUSSION

Page 12, line 523: some comparisons / considerations should be reported with reference to the results of similar research carried out during these more than two years of the COVID-19 pandemic, both in Romania and in other countries.

Authors:

- We have completed the text according to the request: ‘’The negative psychological impact of the COVID-19 pandemic on mothers in the postnatal period was major, with complex manifestations such as depression and anxiety. Romanian studies have shown an increase in the incidence of postpartum depression during the COVID-19 pandemic, with negative consequences on the mother-child relationship [66,67]. Some studies from China reported a 27.43% prevalence of depression and a 33% prevalence of anxiety, with an impact on the whole family and the mother-child pair [68,69]. And studies conducted in Canada during the COVID-19 pandemic indicated increased levels of postpartum depression and anxiety symptoms, with negative effects on mother-child bonding [70].’’ Thank you for recommendations.

LIMITATION OF THE STUDY

It should be noted that the psychometric tools adopted in this study are currently not validated for use in samples of Romanian women.

Authors:

- We have completed the text according to the request: ‘’There are limitations related to the validation of psychometric instruments in Romanian samples. EDPS-R is a scale valid only for antenatal depression (in the third trimester of pregnancy), STAI-Y is valid for Romanian women (but not specifically for the postpartum period) and AAS is not valid for new mothers.”

- We have corrected some editing mistakes in English, we have added some references related to the additions in the text and we have renumbered the references and citations in the text.

Thank you for your recommendations.                                                                                                                                 

Kind regards,

Authors

Reviewer 2 Report (Previous Reviewer 1)

The authors did a good job in reviewing the paper in line with the rewievers' suggestions and indeed the revised manuscript is visibly ameliorated. However, the article still shows several limitations and futher revisions are needed.

Firstly, although the Introduction has been definitely improved, the study's objectives and their rationale are still not clear in some points. Objective III is stated, but it is not detailed in the text because of the reported non-significance of the results. If exploring the association between mother's family type and psychological states postpartum was one of the main aims of the study, results should be reported in detail. Furthermore, the third hypothesis in Objective IV is formulated in terms of statistical analyses conducted rather than associations to be tested and thus its rationale is not clear.

In general, several predictive hypotheses (implying an antecedent-consequent relationship) are investigated, but their foundation appears arbitrary (e.g., it is not clear why depression and anxiety postpartum should predict the mother's adult attachment style and not vice versa).

Some results reported in the abstract (for example, "postpartum depression is predictive of number of days of hospitalization for the mother-child couple 116 (β = .30, p < .001), number of acute neonatal disorders (β = .18, p = .017) ant number of chronic diseases 117 of infants (β = .30, p < .001)") are not mentioned in the full text and the presentation of the regressions' results is still unclear.

In lines 707-713, the interpretation of a "marginally significant" model might be avoided for several methodological reasons (see Pritschet, Powell & Horne, 2016).

As a sidenote, Conbach's alpha coefficients for the State Trait Anxiety Inventory scales are way lower than those reported in other studies (whereas the reported reliability estimates for the other employed instruments are all high and in line with the literatue). I wonder what are the reasons for this discordant finding according to the authors.

Author Response

Dear Reviewer,
Thank you for the review and recommendations.
We will write down the answer after each of your requests.
Comments and Suggestions for Authors
The authors did a good job in reviewing the paper in line with the reviewers suggestions and indeed the
revised manuscript is visibly ameliorated. However, the article still shows several limitations and further
revisions are needed.
Authors: Thank you for the review.
Firstly, although the Introduction has been definitely improved, the study's objectives and their rationale are
still not clear in some points.
Objective III is stated, but it is not detailed in the text because of the reported non-significance of the results.
If exploring the association between mother's family type and psychological states postpartum was one of
the main aims of the study, results should be reported in detail.
Authors:
- Objective III was not one of the main aims of the study, we eliminated it. I also removed the
corresponding paragraph from the Discussion section. We hope he didn't do anything bad.
Furthermore, the third hypothesis in Objective IV is formulated in terms of statistical analyses conducted
rather than associations to be tested and thus its rationale is not clear.
Authors:
- By renumbering, Objective IV is now Objective III.
- We reformulated hypothesis 3, according to the incitements as follows: ‘’3) Maternal
psychological manifestations will predict sharing of the bed with the child, evaluation of the
child, breastfeeding and the type of nutrition the newborn receives during hospitalization.‘’
Thank you for recommendations.
In general, several predictive hypotheses (implying an antecedent-consequent relationship) are investigated,
but their foundation appears arbitrary (e.g., it is not clear why depression and anxiety postpartum should
predict the mother's adult attachment style and not vice versa).
Authors:
- We argued in the Introduction section why depression and anxiety predict attachment style. We
refer to the attachment theory and we have cited studies that have shown this relationship in the
following paragraph: ’’Many studies have demonstrated the links between the mother's adulttype attachment in the postpartum period and high anxiety or postpartum depression [27]. Other
studies have shown that the mother's attachment type can be a risk factor for postpartum
depression and/or anxiety [28,29]. Stable vulnerability factors (for example, personal
psychopathological history of depression, anxiety and psychopathology of other family members,
personality disorder), specific relationship-related life events (for example, break–ups, marriage,
divorce, death, giving birth), changes in underlying perceptions of self and others can produce
changes in adult-type attachment [30].”
Some results reported in the abstract (for example, "postpartum depression is predictive of number of days
of hospitalization for the mother-child couple 116 (β = .30, p < .001), number of acute neonatal disorders (β =
.18, p = .017) ant number of chronic diseases 117 of infants (β = .30, p < .001)") are not mentioned in the full
text and the presentation of the regressions' results is still unclear.
Authors:
- We have eliminated the results reported in the abstract and which are not found in the text and
we wish it to be good.
In lines 707-713, the interpretation of a "marginally significant" model might be avoided for several
methodological reasons (see Pritschet, Powell & Horne, 2016).
Authors:
- We agree with the comment.
- Additionally, we modified the text as follows: ’’The p-value approaches .05 [Χ2
(6) = 11.78, p =
.067, Nagelkerke R2 = .09] and the model was correctly predicted 74.2 % of the cases.’’
- in the Discussions section, we made the following change: ’’Of the predictors used, only the
mother’s depression was a significant predictor. Recommendations for future studies include
further investigation of the studied relationship and further exploration of its possible effect. The
study can be expanded with a larger sample in order to assure a higher statistical significance.“
Thank you for the recommendation.
As a side note, Cronbach’s alpha coefficients for the State Trait Anxiety Inventory scales are way lower than
those reported in other studies (whereas the reported reliability estimates for the other employed instruments
are all high and in line with the literature). I wonder what are the reasons for this discordant finding according
to the authors.
Authors:
- We have added the following in the text to lines 312-315: “Of course, speculations can be made
on cultural factors, reluctancy to declare a state of anxiety, or even the unclear understanding of
some items by the participants, but we did not want to go into too many details, so as not to
divert attention from the purpose the study.”
- Also, we can expect that, in the case of scales with inverted items, the Cronbach's alpha
coefficient will be lower, because mothers may not read the questions carefully, in the special
postnatal situation.
- Additionally, we have corrected some editing mistakes in English, we have added some
references related to the additions in the text and we have renumbered the references and
citations in the text.
- We made changes requested by another reviewer and correctly positioned Table 1.
Thank you for recommendations.
Kind regards,
Authors

Round 2

Reviewer 2 Report (Previous Reviewer 1)

The authors did a good job in reviewing their work according to reviewers' comments and the paper was definitely ameliorated after the last revisions round. In this last version, the background and rationale for the research are thoroughly presented and the results are clearly exposed and discussed, and overall the article can be considered for publication.

This manuscript is a resubmission of an earlier submission. The following is a list of the peer review reports and author responses from that submission.

Round 1

Reviewer 1 Report

The paper investigates the associations between maternal anxiety and depression and several factors, including both attachment and bonding with the newborn and children health-related outcomes (e.g., days of hospitalization, breastfeeding, acute and chronic diseases). The theme is relevant and I appreciated that the authors collected data regarding a considerable sample with full access to clinical records. However, the manuscript presents major drawbacks and I think that it should be extensively revised in order to make it considerable for publication. Both the Introduction and the Discussion sections are confused, with no clear exposition of the study's hypotheses and the relevance of its findings. The rationale behind some passages is difficult to understand and the results' presentation is unclear.

Below I will report in detail my observations regarding some critical points for each section. Further language editing should also be conducted.

- [Introduction] It is not clear whether there is a precise reference for the opening statements concerning mother-infant contact and breastfeeding. Overall, the section appears as a list of disconnected paragraphs. An effort could be made to make it more fluid providing an explicit overview of the link between mothers' mental health, children health outcomes, and mother-child adaptation. Accordingly, the aims of the study could be illustrated more in-depth, stating the authors' specific hypotheses concerning the investigated variables and their relationships.

- [Materials and Methods] Employed measures could be presented in a more detailed fashion. The correct instruments' names should be used (Spielberger's State-Trait Anxiety Inventory and Revised Adult Attachment Scale) and information about their psychometric properties (both in previous research and in the present study) might be reported.

The paragraph dealing with the Revised Adult Attachment Scale seems truncated (line 150).

The paragraph concerning statistical analyses should be extended. Which analyses were performed for which data and with which purposes might be explicitly stated. References for common statistical methods are not necessary (and textbooks or similar might be preferable over online tutorials that could easily become not accessible and are not always thoroughly reviewed).

- [Results] As previously mentioned, the results' presentation is somewhat confused. It is not clear what is the rationale behind the choice to report certain variables in Table 1 whilst excluding others. Different entries refer to different samples and this should be pointed out (for example, birth weight is reported only for premature newborns). Separate spaces might have been dedicated to mothers' characteristics and children characteristics.

Results of psychological questionnaires could be examined in-depth in this section, also reporting percentages of mothers obtaining anxiety and depression scores indicative of certain degrees of severity (mild, severe, etc.).

Abbreviations (e.g., GA, BW) should be used after the full expression has been introduced.

In the paragraph dealing with correlation analyses, it is not clear which criteria were used by the authors to deem the size of the coefficients (very strong or strong). Furthermore, the fact that some variables are continuous and others are categorical is not mentioned, although this may have an impact on the interpretation of the observed coefficients.

Subparagraph 3.2 is hard to follow and could be re-organized. Results concerning attachment might be reported in sequence (both the linear regression models and the multinomial logistic model).

Table 2 header should be corrected. It should also made be explicit that the results reported in this table refer to a hierarchical regression (the subparagraph heading is misleading since it only mentions logistic regression). It is not clear why the authors subsequently inserted depression, state anxiety, and trait anxiety in a three-step model (the order is arbitrary?). A single multiple regression model would have probably been equally informative.

- [Discussion] Same observations reported for the Introduction. Results are not commented and discussed in light of the cited literature, and the latter is sometimes not pertinent (see for example reference 36, line 328). An effort could be made to convey the relevance of the findings and their implications should be addressed extensively. The study's limitations could also be discussed in this section.

As a sidenote, the authors apparently consider the attachment scores obtained with the RAAS as indicative of the mother's attachment behaviors in the interaction with the newborn, despite the fact that - according to what is reported in the Methods section - the scale measures attachment styles in the context of an adult relationship with a partner. Although this information is reasonably relevant and most likely to be related with the mother-infant relationships and other outcomes, this point could be made clearer in the manuscript.

Author Response

Medicina, MDPI

Letter to Reviewer 1

Title: The influence of maternal psychological manifestations on the mother-child couple in two hospital in Timisoara, Romania

Manuscript old ID: medicina-1843768

Manuscript ID: medicina-1979499

Dear Reviewer,

Thank you for the review and for your recommendations.

We will write down the answer after each of your requests.

- [Introduction

Authors:

- To make the text easier to understand, we followed the link between mothers' mental health, children's health outcomes and mother-child adaptation, as you recommended.

- I clarified the purpose of the work, using objectives and hypotheses. We thus restructured this section and added new references.

- Thank you for your recommendations.

- [Materials and Methods

Authors:

- We have presented this column in more detail.

- We corrected the names of the psychological instruments and calculated the internal consistency coefficient Cronbach alpha for each scale (reliability).

- We have restructured the paragraph dealing with the Revised Attachment Scale for Adults.

- We have expanded the section on statistical analysis with explanations and replaced old references with statistics textbooks.

- Thank you.

- [Results]

Authors:

- We have restructured the explanation of Table 1 in the manuscript, in this section and in the introduction and Results.

- We reported scores on postpartum maternal psychological symptoms.

- We used the abbreviations after the full expression was entered.

- We used the Cohen (1988) classification for the Pearson r correlation coefficient and mentioned that some variables are continuous, and others are categorical.

- We restructured Subparagraph 3.2 as two subparagraphs 3.2 and 3.3.

- We corrected the header of Table 2.

- We restructured this section to be easier to understand (depending on the significance of the result).                                                                                                                                                   - Thank you for your recommendations.

- [Discussion]

Authors:

- The results were commented and discussed more extensively, so the entire column was restructured.

- We have added new references and removed some inappropriate references to the text.     

- We discussed the Limitations of the study.

- The mentioned scale measures attachment styles in the romantic relationship.

- We used this scale due to the many similarities between the relationship with the partner and the mother-child relationship.

- We have adapted Conclusions section.

Thank you for your recommendations.

Kind regards,

Authors

Reviewer 2 Report

The manuscript deals with a topic of great theoretical and practical importance.

The current configuration of the scientific article - lacking in many parts - does not allow us to appreciate what the authors probably intended to make known to the international scientific community with their study. In addition to this, the current methodological limitations present in this first version raise some doubts as to the reliability of the reported statements .

In particular, in this first version of the manuscript the scientific contribution and further knowledge that the authors intend to offer to the current vast literature on the subject is not clear.

Furthermore, the same authors repeatedly report the presence or absence of postpartum depression in the women who participated in the research, despite the fact that the instrument used (EPDS) only detects the level of postnatal depressive symptoms and the risk of postpartum depression.

Overall, I propose a new written version of the work starting with some considerations that I report here.

INTRODUCTION

Most of the sentences refer to a single bibliographic citation. I suggest enriching what has been reported by drawing on additional scientific works and increasing the number of bibliographic entries.

Page 2 lines 65-67: insert bibliographic references.

What prompted the authors to carry out this survey?

Report the hypotheses of the study.

MATERIALS AND METHODS

How many women received the proposal to participate in this study?

How many women refused?

Was the sample made up of primiparous and pluriparous?

Who administered the tests?

The conduct of the study coincided with the WHO declaration of a Covid-19 pandemic. How much might this specific condition have affected the collected data?

Argue and report the references relating to the Romanian validation of the EPDS and that of the STAI. For the latter tool, also report the scientific references that argue the possibility that this test can also be administered in the perinatal period and what are the different cut-offs to be adopted. Likewise, report the references relating to the Romanian validation of the READ ADULT ATTACHMENT RATING SCALE.

DISCUSSION

Re-elaborate the reported debate bearing in mind that it is not justified to state that the women in the research are found to have / not have depression and / or anxiety. In light of the tools used, it is in fact necessary to refer only to the presence of any symptom levels of various entities. The EPDS and the STAI do not allow for a diagnosis.

Report the limitations of this study. For example, the scores obtained are all reported by the subjects through the self-compilation of the tools adopted.

Author Response

Medicina, MDPI

Letter to Reviewer 2

Title: The influence of maternal psychological manifestations on the mother-child couple in two hospitals in Timisoara, Romania

Manuscript old ID: medicina-1843768                                                                      Manuscript ID: medicina-1979499

Dear Reviewer,

Thank you for the review and for your recommendations.

We corrected the text in English.

Authors:

- The text of the manuscript was modified according to your recommendations.

INTRODUCTION

Authors:

- We have rewritten this section and added bibliographic references.

- We clarified the purpose of the study by adding study objectives and hypotheses.

MATERIALS AND METHODS

Authors:

- Due to strict hospitalization guidelines during the COVID-19 pandemic (WHO), we were unable to have an accurate record of the number of women invited to participate in this study, nor of those who declined.

- So the number of hospitalized mothers with newborns enrolled in the study was much smaller, and the collection of specific data was clearly limited.

- The sample of mothers consisted of 73 primiparous singleton pregnancies, 52 secondparous singleton pregnancies, 7 mothers with first/second birth twins, and 22 multiples (including one twin and one triplet pregnancy).

- The COVID-19 pandemic has completely changed the conduct of the study.

- The tests used are self-administered under the supervision of a psychologist, guided by the neonatology ward psychologist.

- We have provided a lot of data on the psychological tests used.

DISCUSSION

Authors:

We have restructured this section taking into account your psychology of not referring to the symptoms of postpartum manifestations. Also, as a result of the reformulation of the text, I partially modified the Conclusions column.

Kind regard,

Authors

Author Response

Medicina, MDPI

Letter to Reviewer 3

Title: The influence of maternal psychological manifestations on the mother-child couple in two hospital in Timisoara, Romania

Manuscript old ID: medicina-1843768                                                                                       Manuscript ID: medicina-1979499

Dear Reviewer,

Thank you for the review and for your recommendations.

We corrected the text in English.

Also, the text of the manuscript was modified according to additional recommendations.

Kind regard,

Authors
